# The Quality of Advice Provided by Pharmacists to Patients Taking Direct Oral Anticoagulants: A Mystery Shopper Study

**DOI:** 10.3390/pharmacy8030164

**Published:** 2020-09-03

**Authors:** Jonathon Ertl, Leanne Chalmers, Luke Bereznicki

**Affiliations:** 1North West Regional Hospital, Burnie, TAS 7320, Australia; jonny.ertl@ths.tas.gov.au; 2School of Pharmacy and Biomedical Sciences, Curtin University, Bentley, WA 6102, Australia; leanne.chalmers@curtin.edu.au; 3School of Pharmacy and Pharmacology, University of Tasmania, Hobart, TAS 7000, Australia

**Keywords:** direct oral anticoagulant, patient agent, pharmacist, preferred outcome, community pharmacy

## Abstract

Pharmacists report being less confident in their knowledge of direct acting oral anticoagulants (DOACs) than of vitamin K antagonists, which may influence their ability to detect and manage complications arising from DOAC use. In a mystery shopper study, patient agents were sent into community pharmacies with symptom or product-related requests related to common complications that might arise during treatment with oral anticoagulants, with each visit being assessed for the preferred outcome. Only 10/41 (24.4%) visits resulted in the preferred outcome. A complete history-taking process, obtaining a medical history, patient characteristics and pharmacist involvement were strong predictors of the preferred outcome being achieved. The preferred outcome was not consistently achieved without pharmacist involvement. The potential for strategies that support comprehensive pharmacist involvement in over-the-counter requests should be considered to ensure the provision of optimal care to patients taking high-risk medications such as DOACs.

## 1. Introduction

Oral anticoagulation is required in patients with atrial fibrillation (AF) due to the increased risk of thromboembolic events, mainly stroke [1,2]. Previously, vitamin K antagonists (VKAs) were the only class of oral anticoagulant (OAC) medication available; however, in more recent years direct oral anticoagulants (DOACs) have become available in Australia [3]. DOACs quickly became the first-line therapy for AF once they entered the Australian market in 2013, because of their superior safety and effectiveness [4].

Effective patient education in the use of OACs has been reported to improve outcomes in patients with AF [5,6,7]. Patients taking DOACs have been reported to have sub-optimal knowledge regarding their anticoagulant therapy relative to the level of knowledge observed among patients taking warfarin [7]. Potential explanations for this include fewer healthcare professional interactions and less intensive counselling when commencing the medication [7]. Adverse effects are common in patients taking OACs and, due to knowledge gaps, patients may rely on pharmacists and other healthcare professionals to identify and manage them [7,8]. Pharmacists have reported being less confident in their knowledge of DOACs than VKAs [9]. This might be hypothesised to negatively impact the education provision to patients taking DOACs and potentially lead to a suboptimal detection rate of OAC adverse effects.

Pharmacists promote and ensure the quality use of medicines for their patients [10]. It follows that pharmacists are expected to counsel patients on prescribed medication as well as for over-the-counter (OTC) medications based on the patients’ needs [10,11]. Structured patient education from pharmacists improves patient knowledge and adherence [12]. When presented with an OTC or minor ailment request, pharmacists are encouraged to employ protocols such as WHAT/STOP/GO [13] to assist in ensuring that they obtain relevant information from the patient, such as medical history, current medications, allergies, details of the presenting complaint and any self-management, before making a clinical decision [13].

Pharmacists are in a key position to provide medication knowledge to patients with AF. Interestingly, pharmacists have been observed to take a different approach when counselling patients on DOACs compared to VKAs [9]. They were less likely to use a checklist to ensure all appropriate points were covered [9], and less confident in their knowledge, skills and access to resources on DOACs [14]. They reported lacking confidence in their ability to manage bleeding and bruising as potential adverse effects, and reported more confidence in general areas of management such as the indication of the medication [14]. These studies did not report on pharmacist knowledge of OACs. It is currently unknown whether this impacts the advice they provide to patients. It is important to identify whether pharmacists can effectively identify potential DOAC-related complications and make appropriate recommendations to improve the care of people with conditions requiring an OAC.

The aim of this study was to investigate the quality of advice provided by pharmacy staff when managing scenarios involving DOACs. More specifically, our objectives were to assess the ability of pharmacists to appropriately manage community pharmacy presentations involving DOACs where patient agent actors presented on behalf of patients with symptoms of potential adverse events or drug interactions, and to identify elements of the history-taking process that were associated with the achievement of the preferred outcome of each scenario.

## 2. Materials and Methods

This prospective study utilised mystery shoppers (MSs) to investigate pharmacists’ ability to appropriately manage scenarios involving potential DOAC-related complications. Three MSs were enlisted to act out one of three cases in each participating community pharmacy. Fourth-year Bachelor of Pharmacy students were invited to participate in this project to act as MSs on behalf of older relatives. Each MS received a small incentive for each pharmacy they visited. Each MS underwent a one-hour training session to familiarise themselves with the cases before visiting the pharmacies. The session included role-playing as both the mystery shopper and as the pharmacist. MSs were instructed to practice further over the following week, and were then approved to commence pharmacy visits. Each MS was given a recording device to record the interaction, if the community pharmacy allowed it.

Community pharmacies were sent an invitation letter to participate in the study (*n* = 48). We aimed to recruit ten to 20 participating community pharmacies, based on previous MS research [15,16]. A follow-up phone call was made to each community pharmacy to answer any questions about the study. Each pharmacy received a small monetary gift as well as a de-identified summary of their results relative to other participating pharmacies. The pharmacist in charge or manager was required to sign a consent form on behalf of the pharmacy to participate. They also had to choose whether they would allow recording devices into the pharmacy. Informed consent was obtained from all staff members in each pharmacy, including consent to being recorded. Pharmacies that did not consent to be recorded were still able to participate in this study.

Each case was designed to target an area of a known knowledge deficiency in patients taking DOACs based on previous research [6]. The areas targeted included a potential drug–drug interaction (rivaroxaban and St John’s Wort), a potential adverse effect of dabigatran (dyspepsia), and a potential adverse effect of apixaban (extensive bruising) [7]. Each case was developed through the collaborative effort of the researchers. Once they were completed, the cases were sent to two clinical pharmacy experts for validation, and minor adjustments were based on their recommendations.

Each MS was given one case to act out in each community pharmacy. Therefore, each MS made 14 visits of a total of 42. The MSs were instructed upon their visit to approach the dispensary and start the case irrespective of whether they were talking to a pharmacist or pharmacy assistant. If the pharmacy had consented to being recorded, the MSs began the recording outside the pharmacy and concealed the device on their person. If a product was recommended, the MS made it known that they were a MS at the conclusion of the interaction. No items were purchased during this investigation.

Upon conclusion of the visit, the MS exited the pharmacy and immediately completed a data collection form. The assessment forms had two sections. One section focused on the history-taking undertaken by the pharmacist (or assistant), including, for example, who the patient was, previous actions undertaken to address the health problem, the symptoms they were experiencing and medical and medication histories. The second section focused on the advice and counselling received by the MS, for example, if the pharmacy staff member identified the potential issue and explained it to the MS. Additional information was also documented, such as what product was recommended (if applicable), whether the MS was assisted by a pharmacist or pharmacy assistant or if they referred the patient to a general practitioner (GP). The MS marked each assessment point with either a ‘yes’ or a ‘no’ and recorded additional information that they deemed relevant in an empty section. The data collection form and recording, if applicable, were given to research staff at the earliest possible convenience.

Data collection occurred over a four-week period with regular communication between the MSs and research staff. The MSs were required to communicate with each other to avoid visiting the same pharmacies at similar times. A member of the research team was present at the first visit for each MS and waited outside the pharmacies. This allowed the MSs the opportunity to address any queries or questions about the interaction.

The recordings were compared to the data collection forms for consistency, and then the data were entered into a Microsoft Excel (Microsoft, Redmond, DC, USA) spreadsheet. Each community pharmacy was assigned a study number and any information that could identify the pharmacy was removed and stored separately. The data were subsequently converted to Jamovi (v1.0.7, Jamovi, Sydney, Australia) for analysis. Aspects of history-taking such as medical history, previous medications, age/gender, lifestyle and if they had tried anything for the condition were investigated in all three cases. Each case was then investigated individually for the appropriate counselling points and if the preferred outcome was achieved.

The primary outcome of interest was defined as whether a visit resulted in the preferred outcome being reached (see Table 1). The Fisher’s exact test was performed to investigate whether pharmacist involvement in the interaction, aspects of history-taking and a complete history-taking process were associated with the achievement of the preferred outcome. *P* < 0.05 was considered statistically significant for all analyses.

This study was approved by the Tasmanian Social Sciences Human Research Ethics Committee on the 14 March 2019 (ref no. H0017868). This work was conducted in accordance with the Declaration of Helsinki.

## 3. Results

Of the 48 pharmacies contacted, 45.8% (*n* = 22) expressed interest. Fourteen of these pharmacies consented to participate (69.5%), nine of whom also consented to being recorded (see Figure 1). Each MS conducted 14 visits leading to a total of 42 visits. Data from 41 visits were eligible for analysis, with one visit being excluded due to the pharmacy not stocking the requested product (St John’s Wort). One error was identified between the audio recordings and the MS-reported data in this study, which were corrected based on the audio recording prior to analysis.

### 3.1. Pharmacy Characteristics

Eleven of the 14 pharmacies (78.5%) belonged to a multistate chain, while the other three were local independent pharmacies. Six (42.8%) pharmacies were located in a shopping strip, three (21.4%) were located in a shopping centre or mall and the remaining five (35.7%) pharmacies were stand-alone premises.

### 3.2. Primary Outcome

The preferred outcome was achieved in 24.4% (*n* = 10) of all visits. Table 2 shows the proportion of visits that resulted in the preferred outcome by scenario.

### 3.3. Factors Associated with the Preferred Outcome

The impact of aspects of the history-taking process on the preferred outcome are shown in Table 3. Obtaining the patient’s medication history, medical history, patient characteristics and a complete history-taking process were all significant predictors of the preferred outcome being achieved. The most common reason for the preferred outcome not being attained was failure to obtain all the relevant information (*n* = 34, 82.9% of cases). The patients’ current medications were not requested in 48.8% (*n* = 20) of cases, and thus the pharmacy staff involved were not aware that the patient was taking a DOAC. Of the interactions that did include a request for the patient’s current medications, 52.4% (*n* = 11) of cases failed to reach a preferred outcome.

### 3.4. Interactions

Of the 41 MS visits to the community pharmacies, 38.1% (*n* = 16) involved interaction with a pharmacist, either by direct approach from a pharmacist, referral from the assistant to the pharmacist or through the assistant communicating information to the pharmacist. The remaining 61.0% (*n* = 25) of MS visits only involved pharmacy assistants who handled the interaction entirely by themselves, with no pharmacist involvement. Of the 25 interactions handled by pharmacy assistants alone, only 4.0% (*n* = 1) achieved the preferred outcome. Of the 16 interactions involving a pharmacist, 56.3% (*n* = 9) achieved the preferred outcome. There was a statistically significant association between pharmacist involvement in the interactions and obtaining the preferred outcome (*p* < 0.001).

## 4. Discussion

Only 1 in 4 visits resulted in the preferred outcome. Pharmacist involvement, a complete history-taking process, and enquiring about medication history, medical history and patient characteristics were all associated with the preferred outcome being achieved. Almost 60% of visits with pharmacist involvement achieved the preferred outcome, while the preferred outcome was only achieved in one visit where a pharmacist was not involved.

Overall, this study highlights the importance of pharmacist involvement in presentations to community pharmacies. The underlying issues identified could potentially have result in suboptimal patient outcomes in practice. When investigating factors that contributed to the results, the history-taking process of most visits was suboptimal. A complete history-taking process was conducted in approximately 1 in 6 visits. Many interactions were conducted without the full patient history, with the medication history being requested in just over half of the visits. Therefore, in many cases, the pharmacist or pharmacy assistant conducted the interaction without the knowledge the patient was taking a DOAC. These results highlight the importance of a thorough history-taking process, and the need to be more cautious when managing simple treatment requests. These requests (involving OTC and herbal products) may be seen as involving safe medications, potentially leading to pharmacists and especially pharmacy assistants not taking a thorough history. These results highlight a potential need for pharmacy assistants to receive further training in effectively obtaining clinically relevant information and referring patients to the pharmacist for further consideration when prescription medications are involved.

This is the first prospective study into the quality of education provision provided by pharmacists to patients surrounding DOAC drug interactions, dyspepsia and bruising. Despite being the first study of its kind, many of the findings are consistent with previous research. The results highlight the importance of obtaining an appropriate patient history that is consistent with previous research that demonstrates an association between an increased number of information gathering questions and reaching an appropriate outcome [15,17,18]. The results also highlight the importance of pharmacist involvement in patient interactions, with their involvement being a significant precursor to a preferred outcome being reached. This is consistent with previous research [15,19]. Encouraging pharmacist participation in patient interactions through medication scheduling has been shown to impact the outcome for patients [20], while formal, protocol-driven Minor Ailments Schemes have been demonstrated to improve the efficiency of healthcare delivery [21] and improve clinical and humanistic outcomes [4]. Widespread implementation of a Minor Ailments Scheme in the Australian setting has the potential to support community pharmacists in the provision of consistent, high-quality and appropriately remunerated primary care for all patients, including those taking high-risk prescription medications.

There are other possible explanations for the results apart from suboptimal history-taking. In visits where the medication history was investigated, just over half of the visits still failed to reach a preferred outcome, despite staff being aware of the DOAC. These results may highlight knowledge deficiencies in areas of DOAC management; however, further research is required to investigate this. There was a large variation in the percentage of preferred outcomes achieved between the three cases. The preferred outcome was most commonly achieved in the bruising scenario, followed by the drug interaction scenario and the dyspepsia scenario. Bruising is a side effect of all OACs, whereas the drug interaction and dyspepsia are DOAC-specific complications. Pharmacists have reported being more confident counselling on VKAs [14] and, as bruising is a common side effect of warfarin, pharmacists may be more likely to identify it as a side effect of all OACs. DOACs are also safer medications, potentially leading to fewer investigations into drug interactions. Dyspepsia is an adverse effect of dabigatran, which is the least used DOAC in Australia [22]. Therefore, it may be seldom seen in practice, which may lead to pharmacists overlooking it as the cause.

There are several limitations to this study. Participation in this study was voluntary. Therefore, there may have been a bias in pharmacy selection leading to more confident pharmacies choosing to participate. A small number of pharmacies were involved in the study, and the findings are not necessarily representative of Australian pharmacy practice. Utilisation of MSs is a well-documented approach that avoids many of the biases that plague other methodologies [23]. However, it only assesses a small number of individuals at a specific time. The impact of the business of the pharmacies and the impact of the time spent with the MSs was not taken into consideration when interpreting the results. Furthermore, patient outcomes are not assessed in this methodology. The true impact of the management of these scenarios is unknown. Student MSs have been employed in previous studies to carry out simulated scenarios in community pharmacies [15,16]. However, students have not been validated as accurate and convincing MSs in the same way other groups have been. Previous research has shown that MSs have difficulty maintaining the appropriate standard, leading to some data being excluded [24]. Despite undergoing training, the student MSs may have inconsistently acted out the scenarios. Furthermore, the student MSs may not have been convincing as pseudo-patients. Therefore, the MSs may have been identified more often than reported, potentially influencing the results. Lastly, the recall bias of the MSs must be considered when interpreting the results. Previous research has shown that MSs have an error rate of 10% when reporting data [25]. Five pharmacies out of the 14 did not consent to audio-recording. Out of the pharmacies that did consent to audio recording, some of the audio was poor quality and certain words were hard to understand. Therefore, many interactions relied completely on the MS recalling the interaction.

The scenarios used have not been validated in practice for the purpose of MS visits, which may influence the results. The scenarios may also have influenced the likelihood of a preferred outcome being achieved. Two of the three scenarios were product requests. Previous research has demonstrated that product-request scenarios are performed more poorly than symptom based scenarios [20]. Furthermore, two of the three scenarios required a direct referral to a GP, whilst the third required an instruction to visit a GP if symptoms continued. These scenarios could arguably be more difficult than scenarios that do not require a GP referral, leading to inferior outcomes.

Future studies should look to expand this investigation to a wider range of pharmacies. Studies might also investigate whether pharmacists have knowledge gaps regarding DOAC management, and investigate the potential for improved pharmacy assistant training or a protocol-driven Minor Ailments Scheme to improve preferred outcome rates.

## 5. Conclusions

The preferred outcome of the scenarios investigated was not consistently achieved when a pharmacist was not involved in the interaction. Overall, pharmacy staff often failed to identify the potential cause of the presenting complaint and therefore did not consistently reach the preferred outcome. A complete history-taking process, including medication history, significantly improved the outcome of the scenarios. The potential for strategies that support comprehensive pharmacist involvement in OTC requests, including a formal protocol-driven Minor Ailments Scheme, could be considered by professional organisations and/or pharmacy chains to ensure the provision of optimal care to patients taking high-risk medications such as DOACs.

## Figures and Tables

**Figure 1 pharmacy-08-00164-f001:**
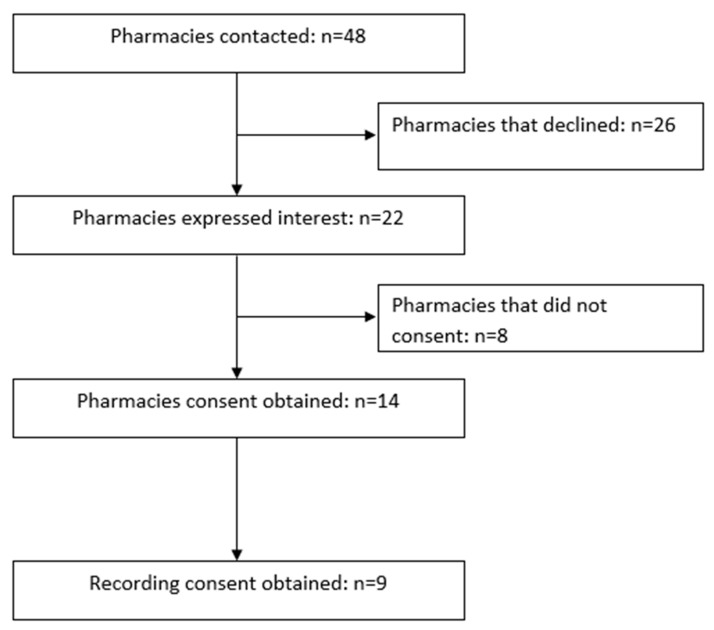
Pharmacy recruitment process.

**Table 1 pharmacy-08-00164-t001:** Summary of mystery shopper (MS) case scenarios, including the preferred outcome for each scenario.

DOAC-Related Problem	Scenario	Patient	Duration of Symptoms	Symptoms	Self-Management	Medical History	Current Medications	Preferred Outcome (All Points Must Be Met)
Dyspepsia	Patient agent presents with a request to treat dyspepsia	55+ year old female	2 weeks	“Heartburn” typically worse at breakfast and dinner No blood in stools or vomiting	Antacid PRN	Atrial fibrillation (diagnosed 2 weeks ago) Hypertension Diabetes	Dabigatran Atenolol Amlodipine Metformin Rosuvastatin	(i) Identification of dabigatran as a potential cause of the symptoms; (ii) Appropriate management advice provided (short-term treatment, lifestyle advice or referral); (iii) Instructions to visit the general practitioner if symptoms persist.
Extensive bruising around body	Patient agent presents requesting arnica cream	75+ year old female	Recent onset	Non-specific bruising across body	No	Atrial fibrillation (diagnosed 1 year ago) Hypertension	Apixaban Metoprolol Amiodarone Rosuvastatin	(i) Identification of apixaban as a potential cause of the bruising; (ii) General practitioner referral.
Rivaroxaban + St John’s Wort interaction	Patient agent presents requesting St John’s Wort	65+ year old male	A couple of months	Visibly down—recommended by friend	No	Atrial fibrillation Cardioversion—4 months ago Hypertension	Rivaroxaban Perindopril	(i) Identification of the potential interaction; (ii) No sale; (iii) General practitioner referral.

**Table 2 pharmacy-08-00164-t002:** Results for preferred outcome by scenario.

Scenario	Visits Eligible for Analysis (*n*)	Percentage (%) of Visits that Reached the Preferred Outcome
Dyspepsia	14	7.14% (*n* = 1)
Bruising	14	50.0% (*n* = 7)
Drug interaction	13	15.3% (*n* = 2)
Total	41	24.4% (*n* = 10)

**Table 3 pharmacy-08-00164-t003:** Associations between aspects of history-taking and achievement of the preferred outcome.

Variable	Number (%) of Interactions Where Relevant History Obtained			*p*
	Total (*n* = 41)	Preferred Outcome Achieved (*n* = 10)	Preferred Outcome not Achieved (*n* = 31)	
Medication history	21 (51.2%)	10 (100%)	11 (35.5%)	<0.001
Medical history	11 (26.8%)	8 (80%)	3 (9.6%)	<0.001
Lifestyle	2 (7.4%) *	0 (0%)	2 (6.5%)	0.60
Patient characteristics (age/gender)	6 (12.2%)	5 (50%)	1 (3.2%)	<0.001
Duration of symptoms	7 (17.1%)	1 (10%)	6 (19.4%)	0.30
Tried anything before	18 (43.9%)	7 (70%)	11 (35.5%)	0.08
Complete history-taking process	7 (17.1%)	6 (60%)	1 (3.2%)	<0.001

* *n* = 27 as it only applied to two cases.

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
