# Peer review of "The Quality of Advice Provided by Pharmacists to Patients Taking Direct Oral Anticoagulants: A Mystery Shopper Study"

_pharmacy, 2020, doi:10.3390/pharmacy8030164_

Round 1

Reviewer 1 Report

This report concerns the real life management of adverse effects of DOACs, when it involves the advice of pharmacists or pharmacy assistants. The conclusions are highly instructive and highlight how critical it is to improve the management knowledge of patients treated  with DOACs on the long-term, and to sensitize on side effects, complications and possible drug interactions (although low with DOACs). Probably, when pharmacists or pharmacy assistants are not confident enough in their advice or knowledge the first recommendation should be to contact rapidly the general practitioner. In any case, when contacting pharmacies, it is mandatory to get the advice of a pharmacist.

In that way, this article presents a high practical interest and it shows how important it is to not reduce the medical follow-up and survey of DOAC treated patients. This study is conducted in Australia, but similar conclusions should apply in most countries. The protocol and the study are well-established and conducted, and limitations are correctly discussed. As participants were aware and volunteer it can be feared that the actual situation is still worse, which can reduce the safety associated to the use of DOACs, although they present many advantages over Vitamin K Antagonists, especially with no or low food interference, and low drug interference. Interestingly, the major drug-specific adverse effect (bruise/bleeding) is better known by pharmacists, than dyspepsia for Dabigatran or St John's Wort drug interference for Rivaroxaban.

A specific comment concerns the abbreviation "DOACs" used: it means "Direct Oral AntiCoagulants" and not "Direct acting Oral AntiCoagulants" (lines 25 and 32). I suggest to remove the term "acting" to comply with current scientific use, although it is a very minor point.

Another specific comment concerns reference 21, which is cited between references 3 and 4. It has probably been included after referencing the others, but for the article format it would be better to review  this referencing to keep the progression of reference numbers.

Author Response

Thank you for your comments. We have made the following changes to the manuscript:

Line

25: Removed – acting in direct acting oral anticoagulant
32: Removed – acting in direct acting oral anticoagulant
34: Reviewed: Reference changed from 21 to 4. All other references adjusted accordingly
133: Added – Date of ethical approval
133: Added – This project followed the rules of the declaration of Helsinki
228: Added – impact of the business/time spent with MSs on results
271: Added – no author received third party financial support

Reviewer 2 Report

In this article from Ertl. et al, the authors present a mistery shopper study, in order to asses pharmacists confidence in detecting and managing complications arising from DOAC use.

The article is really interesting, well written and the methods, the results and the discussion are clearly explained. The overall length of the manuscript is good. However, the number of pharmacy that have accepted to participate in the study is very limited.

The role of the pharmacist as a consultant, as also assessed by this article, is really important. The pharmacy, though, is also a retail shop. Thus, did you take into consideration how busy the pharmacy was and/or the time spent with the MSs comparing it with the outcome?

Author Response

Thank you for your comments.

In this study we did not take into account the busyness of the pharmacy as this would have required additional data collection within the pharmacy that was beyond the scope of the project. We do recognise that this is a limitation, although we aimed to investigate usual processes within community pharmacies, which need to continue to provide quality services, regardless of the busyness of the pharmacy. We have added a statement to this effect in the discussion.